# Cyber Crime Investigation: Landscape, Challenges, and Future Research Directions

Cecelia Horan and Hossein Saiedian *

Electrical Engineering and Computer Science, University of Kansas, Lawrence, KS 66045, USA;
c.horan2018@ku.edu
* Correspondence: saiedian@eecs.ku.edu

**Abstract:** As technology has become pivotal a part of life, it has also become a part of criminal life. Criminals use new technology developments to commit crimes, and investigators must adapt to these changes. Many people have, and will become, victims of cybercrime, making it even more important for investigators to understand current methods used in cyber investigations. The two general categories of cyber investigations are digital forensics and open-source intelligence. Cyber investigations are affecting more than just the investigators. They must determine what tools they need to use based on the information that the tools provide and how effectively the tools and methods work. Tools are any application or device used by investigators, while methods are the process or technique of using a tool. This survey compares the most common methods available to investigators to determine what kind of evidence the methods provide, and which of them are the most effective. To accomplish this, the survey establishes criteria for comparison and conducts an analysis of the tools in both mobile digital forensic and open-source intelligence investigations. We found that there is no single tool or method that can gather all the evidence that investigators require. Many of the tools must be combined to be most effective. However, there are some tools that are more useful than others. Out of all the methods used in mobile digital forensics, logical extraction and hex dumps are the most effective and least likely to cause damage to the data. Among those tools used in open-source intelligence, natural language processing has more applications and uses than any of the other options.

**Keywords:** cybercrime; open-source intelligence; mobile forensics; digital forensics; cyber investigations

## 1. Introduction

Ever since the creation of the Internet, people have been finding ways to conduct illegal activities using it as a tool. In order to counteract these actors, technologies and methods have been developed to track these criminals. It is critical for security and law enforcement professionals to understand these technologies and how they are developing, so they can better perform in their job roles. Internet crime is something that affects anyone who uses a computer, thus making it critically important to counteract it in any way possible.

Some of the most common technologies and methods for tracking cyber criminals are digital forensics and online investigations, which leverages open-source intelligence (OSINT). Within these areas, there are many different technologies and techniques that can be used to gather data on the malicious actor. This data can then be aggregated to determine who committed the crime and build a case against the individual. This paper will cover a survey of these technologies and the methods associated with them.

Digital forensics is a key field used in combating cybercrime because it can be useful if the case is presented in court. Digital forensics helps investigators piece together evidence and determine the timeline of events in a crime. It is mainly made up of network forensics and memory/disk analysis. By analyzing information found on disks and through net-

works, investigators can learn about other potential conspirators in the crime. This could help them track down these individuals and stop them before another crime is committed.

Much of the tracking of criminals is done online. The different layers of crime on the Internet can be broken up into three categories: (1) the surface or open web, (2) the deep web and (3) the dark web. These areas of the web contain a host of information that can be valuable to investigations, so it is important to understand the methods that allow investigators gather and use this information. For example, investigators can utilize information regarding cryptocurrency transactions on the dark web to learn about criminal activity.

The changes and developments in this field are occurring rapidly and it is important for security professionals to keep up to date. Some new developments are coming from automation and machine learning. By automating their tools, investigators can speed up their process and reach their goal sooner. AI forensics will in the future help combat the growing trend of AI crime. This paper will also cover a summary of the developing technologies in this area and how they could change investigations in the future.

Having this information compiled in a single document allows for easy comparison of methods and the information that these methods provide. None of the methods available to investigators are able to gather all the information they require for a case, making it even more important to understand how this information is gathered and how to fill the information gaps. If investigators are able to gain a complete picture of a crime, then they will be able to take action against the criminal or potentially stop a future crime from occurring.

This paper utilizes the basic research and survey methodologies by leveraging existing research, synthesizing the material, and compiling information. Investigators must use a variety of methods, and their knowledge of the field must stay current with any developments. When comparing these technologies and methods, there must be defined criteria of comparison.

When comparing digital forensics methods, there need to be some criteria to compare against. First, the complexity of the method must be determined. This shows how easy it is to perform, how costly it is, and the time consumption of the process. It is important to assess the risks of the technologies and methods used to ensure the integrity of forensic data is kept intact. This paper did not test any of the technologies discussed. Rather, literature on the technologies was analyzed to determine their characteristics.

When comparing methods used in OSINT investigations, these methods must meet some requirements. First, the methods must be faster to use than manually searching for the information. The methods will then be evaluated on their application to the field, namely types of information that can be gathered through each method, the different methods of gathering this information, and the number of different types of cases where this method can be used.

Understanding the methods used to find cyber criminals is an important part of information security because knowing the methods available gives security professionals a deeper understanding of their profession. By understanding these methods and technologies, security professionals can also better understand how crime often occurs, which will help when developing a security plan to prevent crime. Knowing these methods can give deeper understanding because they rely on many of the technologies used in other areas of security, giving security professionals a broader view of their field and the uses of the methods and technologies available.

This paper is intended to compile a summary of technologies and methods used to track criminals online and through forensics, as well as the newest advancements in the field. By organizing this information in one place, it will be easily accessible to anyone interested in knowing about the cutting-edge technologies in this field.

## 2. Digital Forensics

Digital forensics is the practice of collecting and organizing information found on an electronic device for investigative purposes. It is important to know both the technologies and the methods and frameworks investigators use in this field.

Digital forensics can be broken into four areas: host forensics, mobile forensics, network forensics, and cloud forensics. Each of these four areas provides investigators with different kinds of information, with very little overlap. By breaking this field up into these four areas, this paper can analyze the methods for each without covering the same technique twice. This makes it ideal to categorize the methods into these areas because many of the techniques for gathering and analyzing the information from these sources are unique to each source.

### 2.1. Host Forensics

Host forensics is often called digital forensics because it encompasses forensics done on "normal" devices, such as desktops, servers, and other non-specialized sources of data. This method has been long established, but the tools used are constantly evolving as technology is progressing.

Investigators can also utilize the method of weighting forensic evidence with blockchain technology. This can help with certifying the validity of digital evidence with it is presented in a court. This weighting system first collects evidence in a blockchain that records when the evidence was collected and who was in possession of it at the time. This data can then be categorized by relevance to the case and a timeline of events can be created [1]. This method allows investigators to confidently show that evidence was processed correctly and was not tampered with. It can also be helpful with IoT forensics because of the large amount of information gathered in those investigations [2]. An example to demonstrate how weighted forensic evidence can be used is if, after investigators collect evidence for a case, they need some way to prove to the jury that the evidence has not been tampered with. Because of the structure of blockchain and its unchangeable nature, investigators can document the chain of custody for the evidence, showing that it was never unaccounted for.

Something investigators must take into account when performing forensics is the operating system of the device in question. Each operating system performs tasks differently and stores information in different places in the system, which affects all areas of digital forensics [3]. It is critical for investigators to be familiar with the many different types and versions of operating systems in order for them to be able to gather all relevant evidence.

Another challenge that investigators face with host forensics is the randomization of kernel addresses. In order to face this problem, investigators can use four approaches to derandomize this information: brute force code, patched code, unpatched code, and read only kernel data. The brute force method simply scans the entire kernel code. For the patched code option, the kernel must know where to apply patches. The signature from this gives investigators insight into the organization of randomized address locations. The unpatched code signatures come from the code that has been identified as having not been patched. Finally, for read only kernel data, static pointers can help investigators shift data to find offsets, which will lead them to the proper address [4].

### 2.2. Mobile Forensics

As technology has developed, mobile devices have become more common. This means that mobile forensics is a critical part of investigations and should be understood by anyone in the field. Mobile forensics is distinct from any other kind of forensics because of the difference in "hardware, software, power consumption, and overall mobility" [3]. Furthermore, mobile devices are presumed to have personal data, which could be critical to an investigation.

### 2.2.1. Investigation Phases

There are four investigation phases in mobile forensics investigations: preservation, acquisition, examination analysis, and reporting. These phases are depicted in Figure 1.

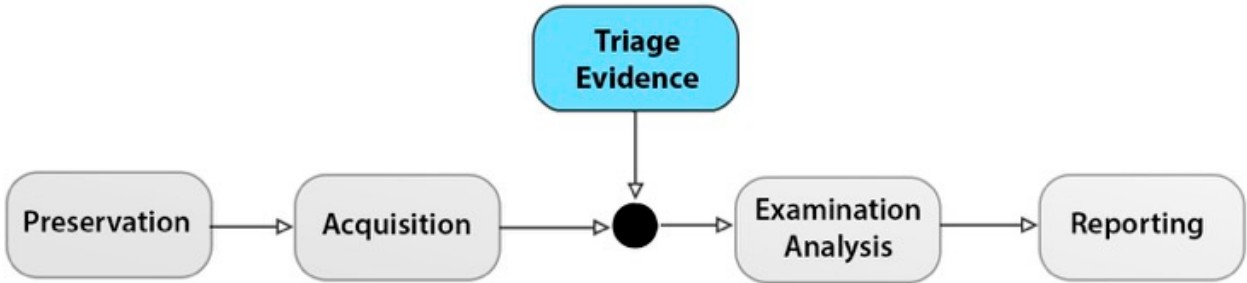

**Figure 1.** Mobile forensic investigative phases [3].

The preservation phase is where mobile devices are taken by investigators and tracked to ensure that the data on them is not tampered with. The acquisition phase is where the data on the mobile device is copied to another device for the analysis that occurs in the examination analysis phase. Finally, the reporting phase is where all the information investigators uncover in the examination analysis phase is documented [3]. Each of these phases must be followed properly to ensure the integrity of any investigation involving mobile devices.

### 2.2.2. Data Extraction

There are common collection methods, also called data extraction, used in mobile forensics. Data from mobile devices must be extracted during investigations. There are five levels of data extraction: manual, logical, hex dumps, chip-offs, and micro reads [5]. Each of these options allow investigators to gather different information from different areas of the device with varying levels of complexity. Table 1 shows a comparison of these methods based on the criteria described in Section 1.

**Table 1.** Comparison of the methods of mobile data extraction [5].

| Method | Complexity | Risk | Notes |
| --- | --- | --- | --- |
| **Manual Extraction** | Low Complexity | High Risk | Puts the integrity of the data at risk of accidental tampering |
| **Logical Extraction** | Low Complexity | Low Risk | Utilizes an external workstation |
| **Hex Dumps** | Medium Complexity | Low Risk | Analyzes dumps of flash memory on an external device |
| **Chip-offs** | High Complexity | Medium Risk | Physically removes the flash memory |
| **Micro Reads** | High Complexity | High Risk | A last resort option because it is very complex and time consuming |

Manual extractions have the lowest complexity because this is where investigators interact with the device using normal methods, such as the touch screen. However, this method can be risky because investigators could accidentally damage or modify the data on the device. It is not advised to use this method because it puts the evidence at risk of destruction or modification, which could make the evidence unusable in the case if it went to court.

Logical extraction is where investigators will extract data from the device to an external workstation using technology such as Bluetooth or a USB. This method also has a risk of inadvertent data modification. Logical extraction is a good method to begin with during an investigation because it allows investigators to analyze data from a different device. It is worth noting that with logical extraction, a pin or password may be required to access the data, which could cause legal issues or complications.

Hex dumps require specialized tools to download the device's flash memory and allows investigators to access data remnants. It is a good way to read and analyze bits of data that may be residing between larger files. This method, however, can be difficult

because it requires investigators to parse memory, which can be challenging and requires specialized training.

Chip-offs are where investigators physically remove flash memory and create a binary image of it that can help in traditional analysis. However, it presents the danger of physical damage to the device, making it medium risk. This is less advised than hex dumps because, as with manual extractions, this puts the evidence at risk of destruction or modification.

Finally, micro reads are the most complicated method out of these five. They use electron microscopes to analyze the logic gates in order to determine the readable data. This method is considered a "last resort" method because it is challenging and resource exhaustive. Micro reads are not applicable to many case scenarios because of their challenging nature.

As shown in Table 1, manual extraction, although easy to perform, is the least recommended because of the risk it poses to the data's integrity. The best methods are logical extraction and hex dumps. These analyze information from different places, so they give investigators a method of gathering different evidence that the other method does not access. Logical extraction and hex dumps have medium or low complexity, making them faster and more efficient to use. Finally, both of these methods pose a low risk to data integrity because they utilize a separate workstation for data manipulation.

*2.3. Network Forensics*

Network forensics is the practice of analyzing information from a host or an entire network [5]. The forensic information can be obtained through logs or traffic captures.

Three of the layers of the TCP/IP Model can provide investigators with useful information. These layers are the application, transport, and network layers. The only layer not included in this is the network interface layer, which includes ethernet frames and the physical connections of a network. Forensic information can be gathered from the application layer through logs that hosts gather. This can be information regarding failed logons or timestamps, which could be critical information in an investigation. The transport and network layer are where firewalls are classified. Firewalls, if properly configured, can contain log data of traffic that has been dropped from the network [6]. This can give investigators information about potentially malicious traffic that has been seen by the firewall. Figure 2 shows the relationship between the layers and the information that investigators can gather by showing the flow of traffic though the network model and the devices it affects.

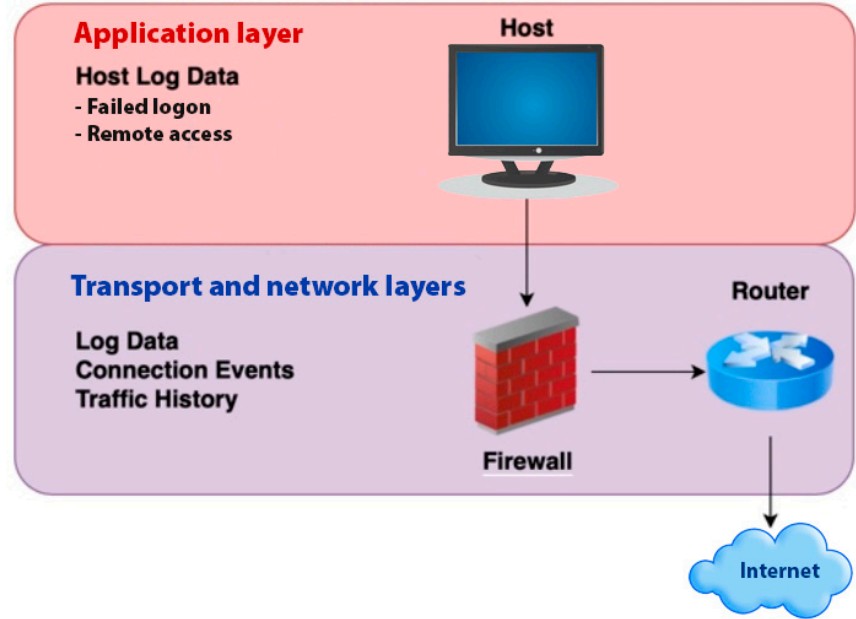

**Figure 2.** Evidence gathered from network layers.

The largest challenge facing investigators in network forensics is the amount of traffic and log data that can be present in an investigation. Although it is possible in theory, it is impossible in practice for investigators to collect and analyze every single packet in a capture of an entire network. The amount of data will not only take too much time to analyze, but also can incur on significant, and often, unmanageable costs [6].

Another challenge that investigators face with network forensics is the growing trend of the Internet of things (IoT). These devices rely on networks to function, meaning there are more end hosts on networks that create logs and traffic. This not only increases the challenge of log and capture size, but it also complicates investigations when determining the scope of the investigation [7].

### 2.4. Cloud Forensics

Cloud forensics is the practice of analyzing data from cloud services and infrastructure in order to gather information for an investigation [8]. Cloud technology is becoming more popular among businesses and individuals. This means that it is a crucial area for investigators to understand. This section discusses the relevant technologies, methods, and frameworks that affect gathering forensic data from cloud sources and using the cloud in forensic investigations.

### 2.4.1. Forensics as a Service

A new development that is changing the field of forensics, especially cloud forensics, is forensics as a service (FaaS). FaaS is a cloud-based service where an organization or individual will pay for the forensics services of another company, similar to cloud computing with providers, such as Amazon's AWS. FaaS is changing how forensics is being handled by moving it further into the cloud, which makes cloud forensics more important to understand [3].

### 2.4.2. Methods and Frameworks

Manral et al., (2020) breaks cloud forensics into two sections, agent-based solutions and log-based solutions. Log forensics are more popular and widely used. These can be spread into four kinds of investigations: incident driven, provider driven, consumer driven, and resource driven investigations. Figure 3 illustrates the differences between these two methods.

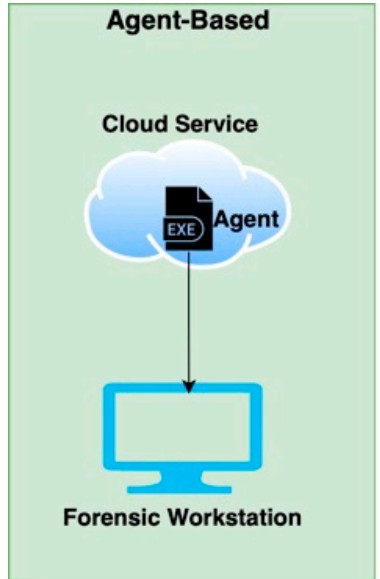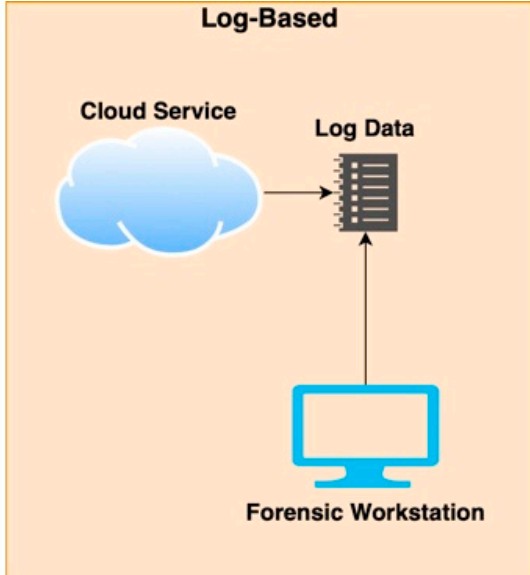

**Figure 3.** Agent-based versus log-based forensics.

Agent-based cloud forensics relies on an application that collects information and sends it back to another location where it can be forensically analyzed. An example of agent-based cloud forensics is having an application, or agent, in the VM being used by the client that sends forensics reports to the investigators [8].

Log-based cloud forensics relies on logs created from events that occur in the cloud that can be then forensically analyzed. As Khan et al., (2016) discussed, cloud log forensics can be broken into three subsections: investigation, synchronization, and security [9]. Investigation is focused on analyzing log files for vulnerabilities that could have been caused inadvertently or through malicious intent. Synchronization is focused on creating consistency across different log files from different sources. Security is focused on keeping log files safe from users that may harm the integrity of the data either inadvertently or on purpose.

### 2.4.3. Cloud Forensics and Mobile Devices

Cloud forensics and mobile devices are treated differently than other cloud and mobile forensic areas. Because of the growing trend to use cloud computing with mobile devices, investigators must account for the cloud aspects of investigations that involve mobile devices [3]. This provides a challenge to investigators because they must account for two different types of forensics during an investigation. One way proposed by Barmpatsalou et al., (2018) is continuous monitoring, where a monitoring system will track and report incidents on the device.

### 3. Online Investigations

Online investigations are the process of gathering, structuring, and using information that can be obtained online. These can be performed by law enforcement, security professionals, or any individual. The main method for gathering information in online investigations is Open-source intelligence (OSINT). This method is the aggregation and use of the information that is gathered using other methods described in sections below. The information gathered for this type of investigation shows relationships, identities, or events that are relevant to the cyber investigation.

### 3.1. Sources of Information

There are many sources of information that investigators use in online investigations. The three main sources are the open, deep, and dark web. Each of these sources can provide investigators with valuable information. Figure 4 shows an illustration of these layers of the web and how the information overlaps. It is important to note that the dark web is a subset of the deep web, not a separate source of information.

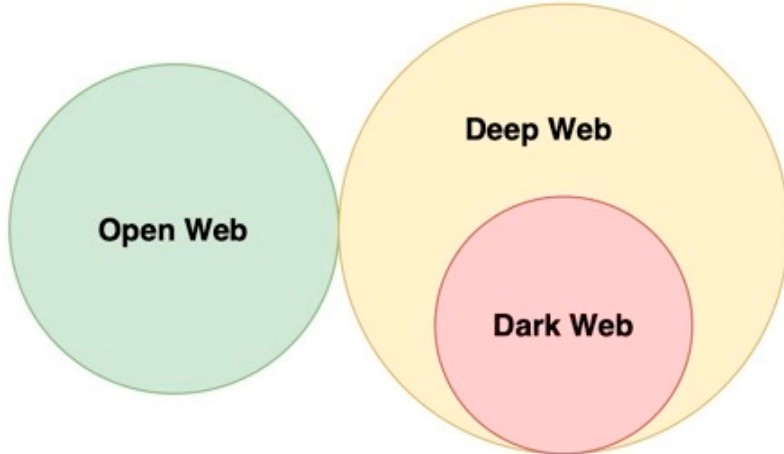

**Figure 4.** Layers of the web.

### 3.1.1. Open Web

The open web is the part of the Internet that is open to all users and indexed by normal search engines, such as Google. This searching method can provide investigators with helpful information. For example, there are some forums and chats that criminals may use that are available on the open web. When this information is collected and analyzed, it can assist in investigations.

There is a wealth of information available on the Internet about crimes and criminal activity. This information can be mined and aggregated to help investigators learn about these activities.

### 3.1.2. Deep Web

The deep web is the part of the Internet that is not indexed by search engines. This is the area of the Internet is not necessarily illegal, but it can be. An example of a deep web site is any site that requires login credentials to access the content [10]. This could be a simple news article or a streaming service that requires payment. The deep web can provide investigators with various types of helpful information. Some of this information could be chat logs that are linked to an individual's account. The information could help identify criminals and their social networks.

### 3.1.3. Dark Web

The dark web is the subset of the deep web where illegal activity occurs. It can be accessed using specialized software, such as Tor, that allows users to access servers, forums, and blogs that would be otherwise unavailable to users. Investigators can also use this specialized software to crawl the dark web for information [10]. The dark web has information that can be useful to investigators in many ways. Nazah et al., (2020) discussed eight major information and crime types on the dark web: human trafficking, pornography, child pornography, assassination, drug selling, terrorist activity, cybercrime markets, and cryptocurrency exchange. This information can be used in investigations to determine the identity, motivation, or even location, of the criminal. It is also helpful for investigators to crawl the dark web in order to learn about new threats, such as new variants of drugs, or new sellers [11]. The lifespans of the marketplaces that criminals use to sell their illegal goods has become shorter [12]. This means that it has become more important for investigators to continuously monitor the activity on these marketplaces to gather as much information as possible before it becomes unavailable.

### 3.2. Specialized Sources of Information

There are specialized sources of information that investigators can gather evidence from: social media and bitcoin flow. These sources can be accessed through the open or deep web, but they offer investigators with specific types of information that can be gathered as evidence. Specialized sources are not separate from the previously mentioned sources. Rather, they are sources that have specific methods attached to them and are often treated differently than other sources.

### 3.2.1. Social Media

Social media is one of the greatest tools that investigators have when it comes to online investigations. Information that is scraped from social media sites or profiles can assist in investigations. It has a wealth of information about organizations and personal lives, not only of regular people, but also criminals and criminal activity as well. For example, investigators use Twitter specifically to identify authors based on URLs, hashtags, replies, and tweet content [13]. As Nazah et al., (2020) stated, "to communicate and sell stolen identities, credit card numbers and other information, cybercriminals rely heavily on social media platforms such as Facebook, Snapchat Instagram, WhatsApp, Telegram and other social media platforms". This means that these data, which could be critical to

an investigation, can exist in the devices used for communication, namely smartphones, or in the cloud as it is the case for WhatsApp's backups.

3.2.2. Cryptocurrency Flow

Cryptocurrencies are often used in cases of ransomware where a criminal is attempting to collect a ransom from their victims. Bitcoin is the most commonly use cryptocurrency by cyber criminals when getting money out of their victims. Cryptocurrency flow is the task of analyzing records of cryptocurrency transactions associated with a crime. It is common for investigators to inspect financial statements in an investigation to determine who is involved in an organization and how a crime was financed. In recent times, many criminals have moved to cryptocurrencies to conduct the financial transactions surrounding the crime. By analyzing the flow of Bitcoin across the dark web, investigators can perform financial analysis. They can sometimes also determine the location of criminals based on transaction records and wallet addresses associated with individuals [14]. This could also be specifically helpful in financial crime cases.

*3.3. Data Mining*

Data mining is the practice of searching the web for information, organizing this information into a report, and using it in an investigation. Edwards et al., (2015) discussed five data mining methods that are used in investigations: natural language processing, information extraction, social network analysis, computer vision, and machine learning. Machine learning will be discussed in section four of this paper. Each of the other methods are described below. Out of the methods described by Edwards et al., (2015), natural language processing and social network analysis were the most commonly used in industry.

As described in Section 1, the methods used in data mining must meet several criteria. First, they must be more efficient for investigators to use. All four of the methods described in Table 2 meet the first criteria of being faster than manual searching. They are compared in Table 2 based on where the methods get their information, how many cases they are applicable to, and how many distinct methods are found.

**Table 2.** Comparison of data mining methods.

| Method | Sources of Information | Number of Cases | Methods of Obtaining Information | Notes |
|---|---|---|---|---|
| Natural Language Processing | 1 | 18 | 85 | Contains four subcategories, each of which can be used in investigations |
| Social Network Analysis | 1 | 5 | 22 | Looks for relationships and patterns in user activity |
| Information Extraction | 4 | 6 | 39 | Utilizes web crawling technology to look for crime trademarks |
| Computer Vision | 3 | 6 | 21 | Searches images, video, and audio for criminal content |

Natural language processing is the most commonly used method, with 85 methods and 18 unique case types available to investigators, even though there is only one source of information. This is largely from the subcategories of this method that can be applied to the same types of cases but yield different types of information that can be used by investigators. For example, authorship profiling can be applied to cases of terrorism and extremism in order to determine attributes of the author, such as militancy, which gives investigators an indication of whether the author is a threat. Sentiment analysis can also be used in cases of terrorism and extremism because it gives investigators the ability to identify the emotions that the author of a text is experiencing, which indicates if the individual is, or is not, part of the criminal organization. The fact that two subcategories can be used on the same case increases the number of uses for natural language processing.

These methods can also affect each other. For example, information extraction often uses natural language processing tools, and sometimes social network analysis tools, in order to create a report on the information found. This still is categorized as an information

extraction example because of the web crawling technology, but it would not be as effective without the use of other methods.

All the methods are applicable to cases in many different ways, as seen in the "methods of obtaining information" column. This shows that there are various ways investigators can gather and analyze the information used in these methods. However, as with the number of applicable cases, natural language processing is clearly the most applicable method for data mining and analysis.

### 3.3.1. Natural Language Processing

Natural language processing is the relationship between human languages and computing. It analyzes one type of information: text. There are four main categories within natural language processing: authorship analysis, author profiling, sentiment analysis, and text classification. Authorship analysis is the process of determining who the author is of a particular text. Author profiling is the process of analyzing a piece of text in order to determine the characteristics of the author. Sentiment analysis is the identification of the motivation behind a piece of text. Finally, text classification is determining where a piece of text falls in various predetermined categories [15]. Natural language processing is by far the most used method out of the ones analyzed in this survey. It only has one information source, text, but it can be applied to cases in many ways.

Authorship attribution is commonly used in online identification. This method allows investigators to determine the identity of the individual behind a piece of text. Authorship attribution can be done by clustering structural, linguistic, or orthographic features that appear in the text. This can help identify previous texts that match the clusters, helping investigators identify the author.

Author profiling is most often related to crimes against children. Author profiling uses common characteristics of language used by certain demographics to determine some characteristics of the author, such as age. Using this method, investigators can detect the ages of the authors, which can identify any mined chat data where one author is a child and the other is an adult. This can be an identifier of a potential crime.

Sentiment analysis can be used for terrorism and extremism cases and harassment cases. In both of these case types, this method is used to detect emotion, the direction of that emotion, and the strength of the emotion. It is able to do this by categorizing the lexicon of the text and determining the tone of the writing.

Finally, text classification can be used in cases of crimes against children. Investigators can determine and then define the key characteristics of child abuse media. Then, they will be able to classify media according to these definitions. One of the most common ways to identify and define these media types is by commonly used keywords.

### 3.3.2. Social Network Analysis

Social network analysis is the use of technologies to learn about the network among criminal organizations and platforms. This method uses tools to scrape information about a criminal or terrorist organization and their connections from an online source [14]. The information for social network analysis is obtained from only one information source, text, but it is still one of the most helpful tools for open-source intelligence investigations.

Investigators have various tools and methods they can use in analyzing social networks. One of the most useful method is scraping data from blog posts and forums that are known to have criminal activity. Another method used by investigators is mining data from graphical information, such as YouTube's social graph. This graph reveals connections between extremist videos and communities.

In social network analysis, it can also be important to employ natural language processing techniques, such as emotion detection [16]. This can help investigators determine the relationship between individuals and groups, helping them know who is associated with the group and their activities.

There are two main areas where social network analysis is used in investigations: terrorism/extremism cases and criminal organization cases [15]. Both of these crime types rely on the identification of organizations and attributing crimes to organizations. Because social network analysis is mainly used to determine the relationships between individuals and entities, it is well suited for these cases because they focus on the structure of the organizations being investigated.

Social network analysis can be used in terrorism and extremism cases when analyzing the activity found on dark web forums of individuals involved in these organizations. This will give investigators the ability to know who is in these organizations, who the key people are, and potentially give them insight into the past and future activity of the organization [14]. A method applied to this area focuses on the process of online radicalization, searching forums based in various locations to determine who the recruiters of an organization are.

When this method is applied to investigations into criminal organizations, the tools and methods are similar to those for terrorism and extremism cases. However, the sources of information are more likely to be news articles and other text-based pages. Investigators can search for terms, names, and geolocation data to learn about the social network that exists between criminals and criminal organizations.

### 3.3.3. Information Extraction

Information extraction is the process of automatically extracting and organizing information. This takes the methods previously described and organizes the information scraped from the web into a report. Information extraction is designed to reduce the time load on investigators because it gathers information and generates a report, reducing the manual efforts of investigators. The four main sources for collecting this information are URLs, technical sophistication, text, and webpages.

The tools used for this method must be able to account for different interfaces, such as online databases. One of the common tools used for this method is web crawlers that will search for any page that is associated with a site and reports on the common topics on these sites [15]. This will give investigators the ability to learn about potential future targets and criminal events. The four main types of information that information extraction utilizes in its analysis are URLs, levels of technical sophistication, text, and webpages. These sources can yield information about the relationships and abilities of individuals and organizations.

Information extraction is commonly used in cases of terrorism and extremism. Forums and websites that are associated with these activities are common sources of information for investigators. Investigators can scrape the forums and websites to learn what the common themes of these groups are.

### 3.3.4. Computer Vision

Computer vision is the practice of using images and videos that can be found online to gain information on a target or crime. This includes not only images, but also text and audio found within a video. This method can provide information such as the identities and affiliations of users online, which can be used in investigations [15]. Computer vision gathers its information from three main sources, which are audio, video, and images.

There are many different techniques that can be implemented in the practice of computer vision. One of the most used technique where computer vision can be applied is in the identification of individuals through the sources of information. When identifying individuals, there are multiple methods available to investigators. One method of doing this is using facial recognition software on avatars that are generated from a photograph. This method is found to be accurate, but only if the user uses their own image to craft the avatar [15].

Another common usage of computer vision is spam filtering and phishing attempts. Many spam emails present their messages as images to avoid spam filters. Phishing emails also heavily rely on images to trick the receiver into thinking the email is legitimate.

However, by using computer vision, these images can be inspected for unwanted content. This same concept can be applied to other types of content, specifically crimes against children, threats and harassment, and terrorism. Information gathered by investigators can be inspected for content that would be considered a crime, such as child pornography or threats made in a video.

The main criminal investigations where computer vision can be used are crimes against children, threats and harassment, and terrorism. When used in investigations involving crimes against children, computer vision can be used to detect images or videos that are suspected to contain child abuse content. In cases of threats and harassment or terrorism, computer vision can be used to identify the faces that appear in content that fall into these categories.

## 4. New Forensic Technologies

There are several new technologies and areas of development in this field. Two of the fastest growing areas are automation and machine learning. Automation is the process of performing a task automatically, without any human intervention. Machine learning, also known as artificial intelligence, is the use of algorithms that can be taught to recognize patterns or objects.

### 4.1. Automation

One area that has experienced development regarding automation is the ability for investigators to automatically detect indicators of crime. Liao et al., (2016) discussed a tool, iACE, that can be used to collect intelligence automatically from multiple sources and compare the relationships of the information gathered [17]. This can be extremely helpful to investigators performing online investigations. It reduces the amount of time searching for relevant information and it helps generate reports with information relationships.

Even though automation is not a new technology, it is currently changing the landscape of this field. It presents challenges for investigators when considering the legal issues of automation [18]. Automation poses many unknowns to many legal systems, such as the "grounds of judgment" in relation to the decisions made by the automated system.

### 4.2. Machine Learning (AI)

4.2.1. Machine Learning as an Investigative Tool

Machine learning has revolutionized criminal tracking and investigations. It can be applied to investigations in various ways. Investigators can use machine learning to teach their systems how to recognize elements of crimes from sources like social media or surveillance footage [19].

This method can be applied to online identification. It uses machine learning to teach the system how to recognize what criminal organization may be behind a crime by what trademarks are seen in the crime. For example, different scamming organizations have different methods of operation. As Edwards et al., (2015) discussed, investigators are able to detect which organizations are behind scams using machine learning techniques.

Another use of machine learning can be in coordination with computer vision [20]. It can be applied to the process of detecting the identity of individuals in videos or images, helping investigators determine the identity of individuals associated with crimes.

Machine learning is most often used in terrorism and extremism cases and harassment cases. In terrorism cases, investigators are able to identify terrorists and terrorist activities by the online footprints of these organizations. For example, information acquired from a data mine of Twitter can be analyzed using machine learning to detect links within unstructured data [15]. In harassment cases, machine learning can be used to detect text-based threats. Edwards et al., (2015) discussed that threats in emails can be detected using a decision-tree algorithm.

Machine learning is also being developed in the field of digital forensics. AI can be used to analyze forensic material, such as network traffic and taught to look for patterns

that indicate malicious activity [21]. This will make investigators task of analysis faster and less costly. It can also be used to examine network forensic data, such as spam activity [22]. The analysis of this can lead to discoveries of network activity trends, which can indicate potential attacks.

Finally, machine learning can be applied to predicting crime hotspots [23]. Machine learning can be applied to historical crime data in order to determine trends and likely future criminal activity. This can be an important tool for investigators when analyzing crime trends, which can lead to discoveries of criminal locations. It can also help investigators be one step ahead of criminals if they are able to predict likely crimes and warn potential victims to put in safeguards in place.

4.2.2. Machine Learning as a Criminal Tool

Machine learning can also be applied to the criminal aspect of investigations. Figure 5 shows a taxonomy for AI crimes. This taxonomy shows how AI can be used by criminals as a tool, but also as a target of their crimes. If criminals can harm a victim's AI systems, it could cause a lot of damage to the victim and their systems. Also, criminals can essentially teach their AI systems to attack the victim's systems, which causes the attack to be faster and more sophisticated than attacks done by individuals [24]. Because this method can be used by criminals, it is critical for investigators to understand this approach and know how to handle it during investigations.

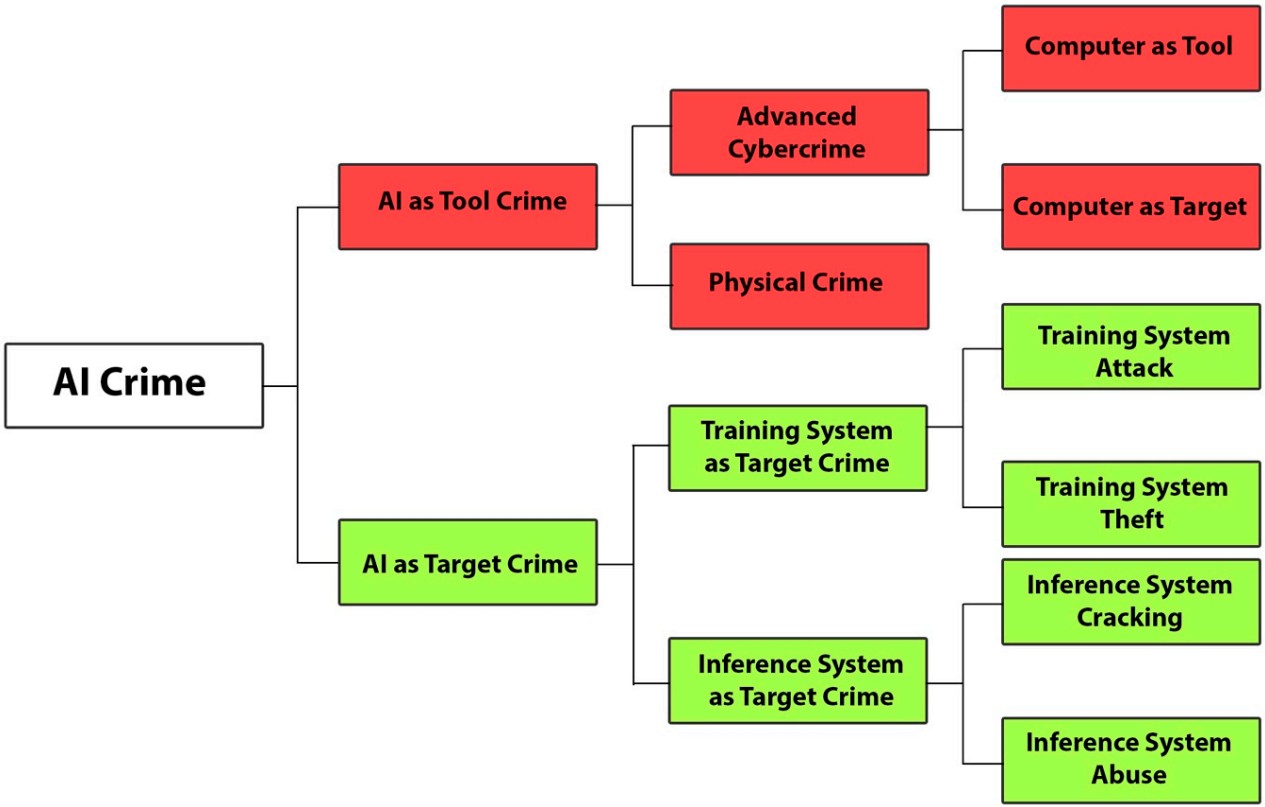

**Figure 5.** AI Crime Taxonomy [24].

## 5. Open Issues and Research Directions

Based on our research, the open issues can be organized into three categories:

- **Technical issues** (e.g., effectively implementing open-source intelligence tools used in investigations).
- **Legal issues** (e.g., obtaining legal basis for collecting evidence that is admissible in courts).

- **Ethical issues** (e.g., criminal profiling).

Each of the above issues will be briefly discussed in this section, and they are depicted in Table 3.

**Table 3.** Open issues in cyber investigations.

| Technical Issues | Legal Issues | Ethical Issues |
|---|---|---|
| Effective implementation | Gathering evidence | Criminal profiling |
| Author identification | Following documented method | Relationships between racial and criminal profiling |
| Big forensic data reduction and management | Chain of custody | Evaluating reliability of criminal profiles |
| Defining data patterns in criminal activities | | Determining the validity of criminal profiles |
| IoT and digital forensics | | |

*5.1. Technical Issues*

Technical issues are categorized as any issue that relates to the technology, tools, or methodology used in the field of cyber investigations. There are a variety of technical issues that present themselves in this field. One is the methodology for implementing tools and techniques in open-source intelligence. This is because each tool provides information differently and there is no well-known "best practice" methodology for investigators to follow. There are many useful tools available to investigators, but there is very little research in this field about deploying these tools. Another technical issue is ensuring the identity of an entity is correct and the proper framework to use when doing this [25]. The goal of performing authorship profiling and attribution is to identify who the author is and who they may be affiliated with. If these processes cannot be verified, the information may become useless in the investigation. Another technical challenge is managing the volume, velocity, variety, and veracity of forensic data stored within desperate data repositories. Some preliminary work is presented by Quick and Choo (2018); they discuss data subsets, digital forensic intelligence and big forensic data reduction and management, but more work is needed. Yet another challenge is to identify and define patterns of criminal or fraudulent activities. Some preliminary work in developing a semantic-based methodology for digital forensics analysis is reported in Amatoa, Castiglione, Cozzolino, and Narducci (2020), but research in this area needs to be further expanded [26]. Given the relevance and the exceeding application and deployment of IoT devices, reliable direction is needed on how to legally and comprehensively collect data from these devices, especially during a critical time. Device manufacturers, researchers, and legal experts should collaborate to address the concerns. Watson and Dehghantanha (2016) discuss the topics, challenges, and the importance of further development in this area [27].

*5.2. Legal Issues*

Legal issues are categorized as any issue investigators might be concerned with regarding the law or presenting the investigation in court. One issue in this category is the topic of maintaining and proving the integrity of digital evidence. Once the forensic evidence has been collected and the case potentially goes to court, investigators will have to present the evidence and establish that it was collected legally. If they cannot do this, the evidence may become inadmissible. Also, another legal issue that investigators may face is following a documented and scientific method to ensure that the evidence can be presented, and even repeated, in court. Finally, investigators must prove a chain of custody of the evidence to ensure that it was not tampered with or accidently modified [28]. This will ensure the integrity and admissibility of the data.

Another legal issue that is present in cyber investigations is the differences between jurisdictions and geographical methodologies. Cybercrime is a global issue and often spans multiple jurisdictions and geographical locations. This means that multiple countries can be affected, which means that law enforcement from these countries must collaborate to investigate the crime properly. The affected countries also often have their own methodologies, tools, and techniques they use in investigations. These differences are an open area of study in this field.

*5.3. Ethical Issues*

Ethical issues are categorized as any moral or ethical dilemma that investigators might have to face in the course of a cyber investigation. For example, in open-source intelligence investigations, one of the techniques that investigators use is profiling based on the information found about an individual or group online. Criminal profiling is used to define a process "in which the nature of a crime is used to make inferences about the personality and other characteristics of the likely offender" [29]. Criminal profiling is used in many types of investigations, and they can lead to problems with biases and assumptions if the tools are not implemented properly and the investigation is not well documented [30]. While a variety of criminal profiling have appeared in media and movies, the validity and reliability of such profiles need to be investigated.

*5.4. Research Directions of Open Issues*

These open issues can lead into interesting research directions. One of these is how to better utilize the technologies used in open-source intelligence as an entire program, and not just individual tools. The tools discussed do not provide information that describes the whole picture for investigators. Researching the best methods of using these tools with each other can help investigators use these tools efficiently during investigations. This research will give investigators a great advantage in their investigations because the conclusions of the research may be able to speed up the investigative process.

This survey can also lead to research in the area of bias handling and equitable treatment in open-source intelligence investigations. These types of investigations are an important part of the whole investigative process, so it is critical that they can be implemented correctly and with fair treatment. This can also help investigators find the truth faster if there are no biases or assumptions built into the methods or tools they use.

Finally, research into legal issues described can help security professionals better perform their jobs. Because criminal investigations can end up in court, it is critical to understand how laws affect the collection and preservation of evidence. This research may not be done by technical professionals, but it is still critical for the technical professionals to understand the non-technical legal issues.

## 6. Conclusions and Further Research

With the advance of technology, criminal investigations rely more on technology as threat actors are becoming more advanced. There are many tools and methods investigators can implement to assist in their jobs. None of these tools can perform all the functions that investigators require. Therefore, it is necessary to become familiar with different tools, how they function, and what information they can provide to investigators.

The field of digital forensics has been around for some time, but it is still evolving. The four main areas, host, mobile, network, and cloud forensics are still critical to digital investigations, especially as new methods are being developed. There are multiple ways to extract data in mobile forensics. The most effective methods are logical extraction and hex dumps. Digital forensics' main area of growth is in the field of cloud forensics, especially as many services are becoming cloud based.

Open-source intelligence and online investigations are not a new method, but investigators are always applying new technologies to these methods. Multiple methods in this field can be applied to online investigations in order to gain as much intelligence on threat actors as possible. Each of these methods provide some information, but none of them provide all the information necessary for an investigation. However, out of all the methods available to investigators, one stands out as having the most applications: natural language processing. This method can be applied to many different cases and provides investigators with several different kinds of information for these cases.

Automation and machine learning are advancing the field technology and cyber investigations are also being affected by these technologies. Automation is helping investigators speed up the process of collecting evidence, while machine learning is helping investigators

identify and classify this evidence. Automation also presents challenges to this field in regard to legal assumptions and implications [18].

This is a growing field and there are many opportunities for further research. Automation and machine learning provide potential areas of further research as these technologies become more sophisticated. Open-source intelligence techniques were also found to be underrepresented in the research field, opening another area for future research.

**Author Contributions:** Conceptualization, C.H. and H.S.; methodology, C.H. and H.S.; formal analysis: H.S.; investigation, C.H.; writing-original draft, C.H., writing-review and editing, C.H. and H.S.; visualization, C.H. and H.S.; project administration H.S. All authors have read and agreed to the published version of the manuscript.

**Funding:** This research received no external funding.

**Institutional Review Board Statement:** Not applicable.

**Informed Consent Statement:** Not applicable.

**Conflicts of Interest:** The authors declare no conflict of interest.

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
