# Peer review of "Cyber Crime Investigation: Landscape, Challenges, and Future Research Directions"

_jcp, doi:10.3390/jcp1040029_

Round 1

Reviewer 1 Report

The authors have addressed all the previously highlighted issues.

There is only an issue with section 3.2.2.:

Section 3.2.2 - Bitcoin flow. This section should be named "Cryptocurrency flow" with an introductory sentence explaining that i) Cryptocurrencies have been instrumental to allow cybercrime to collect ransoms and that ii) Bitcoin is the main cryptocurrency used in cybercrime, although it does not guarantee full anonymity. 
Then, the section can proceed with the existing text about Bitcoin.

Table 2: the heading row of the table is still not visible (this is a publishing issue). Table 3 possibly still has the same issue (not visible in the newly submitted version, as the row is highlighted with yellow color).

Author Response

Please see attached responses.

Reviewer 2 Report

THe paper has been slightly improved, partially following the suggestions. THe novel 5.2 section partially fills the gap identified in the previous review.

FInally, the novel references should be mentioned in the paper, not only listed in the References section.

Author Response

Please see attached responses.

This manuscript is a resubmission of an earlier submission. The following is a list of the peer review reports and author responses from that submission.

Round 1

Reviewer 1 Report

The presented article “Cyber Crime Investigation: Landscape, Challenges, and Future Research Directions” is a survey about the field of digital forensics; therefore, the title might be misleading since many crimes are involved in this topic that are not committed in cyberspace. The paper introduces different digital forensics forms, applied technologies and methodologies, and new trends and approaches. Furthermore, the authors address challenges and identify research gaps and directions.

The topic of this paper is exciting, and the article also makes the reader curious while it presents the related field in a way that is easy to understand. However, this is also the major drawback of the paper since it is more like a popular science paper than an academic one regarding its style and depth.

The paper covers all the important technologies and methods related to digital forensics, and the comparison tables highlight the key parts. Moreover, ethical and legal issues are also presented in the paper so that the reader can interpret the cited solutions from another context.

Nonetheless, the discussed parts are not detailed enough. In most cases, the technical aspects are not revealed, only how they are applied within a specific topic. Of course, this paper focuses on giving a comprehensive survey, but the proposed research gaps have to be more specific by referring to technical details.

To summarize, I liked the paper, and it has the potential to be significant; however, in this form, it lacks a more technical aspect that is fundamental to narrow the identified research gaps – or at least to name some technologies related to the current research gaps

Author Response

Please see attached response to the reviewers.

Reviewer 2 Report

Authors claim to propose a survey which compares the most common tools and methods available in cybercrime forensics. It should determine the information provide by those tools, and which are the most effective.

The paper is not well organized, as its content does not have a survey, but instead it describes the fundamentals behind some topics related with cybercrime and, in general, the tools used to track cybercriminal activities. Despite the relevance of the subjects, the paper is not easy to follow, it has some major methodological flaws and a substantial reformulation should be made.

1) The english could be improved, especially in the clarification of some issues described below. 

2) The bibliographic list should be improved and extended. In a survey a list of up-to-date bibliographic references is expected and in this paper the list is short and with references from 2015, 2018. An improvement is required.

3) The authors does not clearly indicate what are the purposes of this paper. Lines 15-18 should better clarify what are the purposes: to compare tools? In terms of what? what means "... to determine what information the tools and methods provide"? By the way, what is the difference between a tool and a method?
In line 17 you mention a "criteria for comparison", but it is too vague. Section 1.2 vaguely describes some characteristics that methods and tools should have, but not in terms of the methodology adopted. For instance, it is quantitative, qualitative? Did you test the tools? How? What results?
This is a major about the paper: there is no evidence regarding the comparison of the tools and methods.

4) The text on sections 1.1, 1.2. and 1.3. can be written without having the sections.

5) Some phrases need clarification and be supported by bibliographic references. Some examples:
- Line 40: "Digital forensics is a key field to tracking cyber criminals...". Digital forensics is mainly used to look for digital evidences in seized devices and not directly used to "track" cybercriminals. Some clarification is needed or a biblio reference.
- "Internet" instead "internet", throughout the text.
- Line 59: It is correct that there is no method/tool that fit all the investigators' demands. But the paper does not describe tools (commercial or not) and their features.
- Lines 87-90: too vague.
- Lines 95-99: clarification is needed. Why these four areas? A biblio reference could support this paragraph.
- Line 105: what is "weighting forensic evidence with block-105 chain technology"? You need to use blockchain the preserve the chain custody? Can you present an example? 

5) Sections 2.2 and 2.3 does not present novelty and is based in one reference. Why Chip-offs is of Medium Complexity and Medium Risk? It is usually highly complex and rare and with low impact. 

6) Section 2.4. is also too vague. Which methods and tools were analysed?

7) Section 3 has several flaws and some phrases have to be clarified. A concern is the fact that authors add more complexity, by going into open/deep/dark web, besides the description made in Section 2. 

8) Section 3.2. - only these two are "specialized" sources of information? A biblio reference should be used.

9) Table 2 is not correct. It cannot be analysed correctly.

10) Lines 300-301: what are the "fifteen relevant uses in cases"?

11) Line 390: Why these areas? "terrorism and extremism and criminal organization". Based on which authors? It is possible to find additional relevant areas related with social network analysis.

12) Lines 442-443: ??

13) In section 4 you mention automation is a "new" forensic technology, but with a biblio reference of 2016. The same for section 4.2. Perhaps an existing survey of ML as cybercrime tools could help.

14) Table 3 and section 5 should be better framed in the text. Some of the entries are not necessarily "open issues".

Summing up, in my opinion the paper has several flaws that and a deep rewritten should be made. It seems that the authors tried to correlate everything, but a deeper and organized analysis should be made. Regarding tools and methods, the comparison is not grounded on experiments or say quantitative analysis. I'd suggest to define a comparative scale to classify the tools and methods and define a score for each one. For instance for tools related with (host) digital forensics in a first stage.

Author Response

(The authors gave the same response as above.)

Reviewer 3 Report

The paper aims to survey technologies and methods for (mobile) digital
forensics and open source intelligence (OSINT).

The paper tries to cover too much: digital forensics and OSINT. Each of this
subject would be enough for a paper on its own, as these are two wide areas. This results in a paper that are often too vague (on both subjects) to be useful.

Another weakness of the paper lies in the fact that although it uses the  designation "tool", it effectively does not focus on tools, but instead on some technologies and methods. 

Additionally, although the authors use the word "survey" (twice in the
abstract, and two more times in the text), the paper is not a survey, but an
overview of 1) digital forensics (mostly focused on mobile and cloud) and of
2) OSINT. 

Finally, the paper mentions cybercrime, but many of the examples are focused on cyberterrorism. 

Positive points are the OSINT part and IA in digital forensics/OSINT, namely as there is not much literature focusing on OSINT applied to digital forensics.

Some specific comments on the text follow.

Line 40: "Digital forensics is a key field to tracking cyber criminals and counteracting crime. It is mainly made up of network forensics and memory
analysis."
=> This is a bold statement, leaving out analysis of persistent storage (disks,etc.). And then, in the next sentence, authors reference disks
("...found on disks...").

Line 83: "The risk to the data must then be determined, showing the risk to
the data when using the method."
=> Confuse sentence. Suggestion:
"It is important to assess the potential risks of technologies and methods
regarding the integrity of the forensic data. "

Line 139: "A challenge that investigators face with host forensics is the
randomization of kernel..."

=> Is your reference to kernel address randomization linked to memory
forensics? This should be made clear. Additionally, you first write about
"kernels" and then in the next paragraph about operating systems. This should be the other way around (going from generic to specific).

Line 152: "...many different **types** of operating systems..."
=> types and versions

Line 197: "Logical extractions..."
=> In password/pin/passcode protected devices, logical extraction might
require the access code or a tool that can circumvent authentication. This
should be stated in the text.

Line 258: "..., but it also poses investigators with a monetary cost..."

=> "..., but also can incur on significant, and often, not manageable costs"

Line 368: "This means that all this information, which could be critical to an
investigation, is stored with these companies and can potentially be accessed by investigators."

=> WhatsApp is known to use, by default, end-to-end encryption (other IM tools also do the same). Thus, even if WhatsApp's company (Facebook) stores conversation data, these data are encrypted.

Suggestion for the sentence:
"This means that these data, which could be critical to an investigation, can
exist in the devices used for communication, namely smartphones, or in the
cloud as it is the case for WhatsApp's backups".

Line 563: "One method of doing this is using facial recognition software on
avatars that are generated from a photograph. This method is found to be
accurate, but only if the user uses their own image to craft the avatar."

=> This (interesting) sentence should be supported by one or more bibliographic references.

Line 567:"...spam filtering"

You should mention "phishing attempts" to reinforce the mention to SPAM filters. 

Line 592: "It will reduce the amount of time searching for relevant
information and it will help generate reports with information relationships."
=> The use of the future tense ("will") is weird, as the reference for the
iACE tool given by the authors is from 2016.

Line 675: 5.OPEN ISSUES AND RESEARCH DIRECTIONS

Suggestion: Reorganize Section 5 in the following way:
5.1 Technical issues

5.2 Legal issues

5.3 Ethical issues

Writing issues
==============
Abstract: "...the survey establishes criteria for comparison and conducts an
analysis of the tools in both digital forensic and open-source intelligence
investigations." 

=> of the tools in both **mobile** digital forensic and... 

Line 32, line 83, line 148, line 214, line 584, line 592, line 691...: preform => perform

Line 35: "open-source intelligence, or OSINT" 
=> open-source intelligence (OSINT)

Line 56: "...the surface, or open, web, the deep web, and the dark web."
=> : 1) the surface or open web, 2) the deep web and 3) the dark web.

Line 354: "and currency exchange (Bitcoin)"
=> "and cryptocurrency exchange"

Line 513: the heading row of the table is not visible
(the text can be selected, but in the PDF it is not visible -- tested on two
different PDF viewers)

Line 514: "There are two main areas where social network analysis is used in
investigations: terrorism and extremism and criminal organization"
=> The way it is written, one could think that there **three** main areas.
Suggestion: "There are two main areas where social network analysis is used in
investigations: **terrorism/extremism** and criminal organization"

Line 579: "transaction records and wallet addresses associated with
individuals (Nazah et al., 2020)."
=> this looks like lost text.

Line 583: "...to detect **indicators of automatically**"
=> "to automatically detect crime(?) indicators"

Line 689: "about the deploying these tools"
=> "about deploying these tools" 

Line 747: Table 3 is not referenced in the text. Additionally, it should be placed in Section 5.

Author Response

Please see attached document (under Response to Reviewer #1)

Reviewer 4 Report

The paper is clearly written and compares the most common methods available to investigators to determine what kind of evidence the methods provide, and which of them are the most effective. It is an "horizontal" paper, because it crosses different disciplines (IT, ethics, law, organization), whose target is to aim to organize the criminal investigation area, in terms of IT tools effectiveness and related methodologies, analysing the impacts on legal and ethics perimeters for general investigations (Serious Organised Crime has been treated as common crime although the legal/ethical perimeters are different). The paper doesn't show any scientific methodology because it is a survey of the current literature, with no novelties in the presented results. For this reason, although the paper offers a broad perspective on the topic, I believe that it could be improved in terms of geographical results and techniques usage, from quantitative standpoint, therefore having a comparative approach between different usages of IT tools in countries having a different legal/ethical framework. Or analyse differences in IT tools, ethics and law between SOC and common crime scenarios. In brief, the paper mixes pillars in IT Criminal investigations when the requirements are different and have a severe impact on techniques (e.g IT Investigations in Europe are different from Far East).

In page 8, the authors present a definiton of deep web which includes "normal search engines". It would be better to clarify to the reader why a "normal search engine" is different from a "not-normal search engine" (aka, is duck-duck-go on surface and Tor normal or not normal?).

Finally the bibliography could be enriched adding Europol-EC3 threat assements (SOCTA/IOCTA) and some more paper related to specific methodologies (e.g. Tor marketplaces exploratory data analysis: the drugs case, A Celestini, G Me, M Mignone) .

For all the abovementioned reasons, I recommend to accept iff room.

Author Response

Please see the attached document (under Response to Reviewer #2)

Round 2

Reviewer 2 Report

Despite the efforts made by the authors in this second version, some concerns remain, mainly those related with the scientific soundness of this paper and how it can help the cybercrime investigators and practitioners.

1) The comparative study of the methods used for mobile forensics and online investigation (different from OSINT) is not grounded on literature or experiments. Table 2 is more readable now, but needs additional clarifications. For example, what are the "85 methods 391 and 18 unique case types available to investigators" (lines 391-392) in NLP? Or at least a reference to support this statement. The same for the other methods presented. The reader (e.g. cyber investigators) has to have this info. At leat, a description of some of them.
2) line 543 and 560: machine learning is different from AI! Section 4.2.1 has some misconfusion between these two terms.
3) Regarding NLP, can you describe how militancy can be extracted and be used to identify a potential author of terrorism? What is "Authorship profiling"? It is too vague and unclear to the reader.
4) The "number of cases" and "methods of obtaining information" is too vague and not easy to depict from the text in the paper. 
5) I don't understand the change of the bibliographic list style. The style adopted in the first version is the one used by the journal.

The work is relevant and a survey like this is worth for the cybercrime investigators. However, the paper is not well organized and written. There are relevant terms and topics that need clarification and examples, to guide the reader.
I understand that the aim is not the comparison of tools. However, even being a comparison based on the literature, more work should be made to better elucidate the readers of the journal.

Author Response

The attached Word file entitled jcp-1250497-response-to-reviewers-round2.docx contains our response to the reviewers.

Round 3

Reviewer 2 Report

The reviewer's concerns about this paper remains. 
The contributions of this paper are still unclear and several explanations are missing. Tables 1 and 2 is where some comparison is made. Regarding Table 2, I maintain my doubts and concerns, as there are no comparison at all. This table is too vague and, as is, the values that are presented do not tell valuable information to the reader. 

My suggestion to the authors is to have a more dedicated and oriented survey, for example only for NLP or social network analysis, but in depth. For example, what are the 85 methods of obtaining information? Or the 22 for social network analysis? What are the number of "cases" for each method? Where they are/were used? For which purposes? 
Which works, authors, applications and frameworks use each one of the surveyed topics?
An so forth. As it is, is just a set of numbers and with no relevance for the reader, I'm afraid. 

In table 3, "IoT and digital forensics" does not have legal and ethical issues?

Sections 4 and 5 should be revised and reorganized. Some of the topics are not open issues and machine learning is not a "new forensic technology".

Paper should be rewritten and a more concise organization about the contents to be included should also be made. Eventually authors could try to have a survey in one (or two) of the several issues approached in this paper.